# Advancements in Neuroendocrine Neoplasms: Imaging and Future Frontiers

**DOI:** 10.3390/jcm13113281

**Published:** 2024-06-02

**Authors:** Luigi Asmundo, Francesco Rizzetto, Michael Blake, Mark Anderson, Amirkasra Mojtahed, William Bradley, Anuradha Shenoy-Bhangle, Carlos Fernandez-del Castillo, Motaz Qadan, Cristina Ferrone, Jeffrey Clark, Valentina Ambrosini, Maria Picchio, Paola Mapelli, Laura Evangelista, Doris Leithner, Konstantin Nikolaou, Stephan Ursprung, Stefano Fanti, Angelo Vanzulli, Onofrio Antonio Catalano

**Affiliations:** 1Postgraduation School in Radiodiagnostics, Università degli Studi di Milano, Via Festa del Perdono 7, 20122 Milan, Italy; luigi.asmundo@unimi.it; 2Department of Radiology, Harvard Medical School, Massachusetts General Hospital, 55 Fruit Street, Boston, MA 02114, USA; mblake2@mgh.harvard.edu (M.B.); mark.anderson@mgh.harvard.edu (M.A.); amojtahed@mgh.harvard.edu (A.M.); william.bradley@mgh.harvard.edu (W.B.); ashenoy-bhangle@mgh.harvard.edu (A.S.-B.); 3Department of Radiology, ASST Grande Ospedale Metropolitano Niguarda, Piazza Ospedale Maggiore 3, 20162 Milan, Italy; angelo.vanzulli@unimi.it; 4Department of Surgery, Harvard Medical School, Massachusetts General Hospital, 55 Fruit Street, Boston, MA 02114, USA; cfernandez@mgh.harvard.edu (C.F.-d.C.); mqadan@mgh.harvard.edu (M.Q.); 5Department of Surgery, Cedars-Sinai Medical Center, 8700 Beverly Boulevard, Los Angeles, CA 90048, USA; cristina.ferrone@cshs.org; 6Department of Oncology, Harvard Medical School, Massachusetts General Hospital, 55 Fruit Street, Boston, MA 02114, USA; clark.jeffrey@mgh.harvard.edu; 7Nuclear Medicine, IRCCS Azienda Ospedaliero-Universitaria di Bologna, Via Massarenti 9, 40138 Bologna, Italy; valentina.ambrosini@unibo.it (V.A.); stefano.fanti@aosp.bo.it (S.F.); 8Nuclear Medicine, Alma Mater Studiorum University of Bologna, 40126 Bologna, Italy; 9Department of Nuclear Medicine, IRCCS San Raffaele Scientific Institute, Vita-Salute San Raffaele University, 20132 Milan, Italy; picchio.maria@hsr.it (M.P.); paola.mapelli@hsr.it (P.M.); 10Department of Biomedical Sciences, Humanitas University, Via Rita Levi Montalcini 4, Pieve Emanuele, 20072 Milan, Italy; laura.evangelista@hunimed.eu; 11Department of Diagnostic and Interventional Radiology, University Hospital Frankfurt, Theodor-Stern-Kai 7, 60596 Frankfurt am Main, Germany; doris.leithner@nyulangone.org; 12Department of Radiology, University Hospital Tuebingen, Osianderstraße 5, 72076 Tübingen, Germany; konstantin.nikolaou@med.uni-tuebingen.de (K.N.); stephan.ursprung@med.uni-tuebingen.de (S.U.); 13Department of Oncology and Hemato-Oncology, Università Degli Studi di Milano, Via Festa del Perdono 7, 20122 Milan, Italy

**Keywords:** neuroendocrine neoplasms, response criteria, CT, MRI, PET/CT, PET/MRI

## Abstract

Neuroendocrine neoplasms (NENs) are a diverse group of tumors with varying clinical behaviors. Their incidence has risen due to increased awareness, improved diagnostics, and aging populations. The 2019 World Health Organization classification emphasizes integrating radiology and histopathology to characterize NENs and create personalized treatment plans. Imaging methods like CT, MRI, and PET/CT are crucial for detection, staging, treatment planning, and monitoring, but each of them poses different interpretative challenges and none are immune to pitfalls. Treatment options include surgery, targeted therapies, and chemotherapy, based on the tumor type, stage, and patient-specific factors. This review aims to provide insights into the latest developments and challenges in NEN imaging, diagnosis, and management.

## 1. Introduction

Neuroendocrine neoplasms (NENs) constitute a heterogeneous array of neoplasms that express neuroendocrine markers. They originate from different sites within the body, encompassing areas such as the respiratory system and the gastrointestinal tract [1]. Gastroenteropancreatic NENs (GEP-NENs) can be classified into distinct subtypes: well-differentiated gastroenteropancreatic neuroendocrine tumors (GEP-NETs), poorly differentiated gastroenteropancreatic neuroendocrine carcinomas (GEP-NECs), and mixed neuroendocrine/nonendocrine neoplasms (MiNENs) [2]. Similarly, lung NENs (Lu-NENs) are classified as lung neuroendocrine tumors (Lu-NETs) and lung neuroendocrine carcinomas (Lu-NECs), which include typical carcinoids (TCs) and atypical carcinoids (ACs), small cell lung carcinomas (SCLCs), and large cell neuroendocrine carcinomas (LCNECs) [2].

The incidence of NENs has significantly increased in recent decades, at least in part due to greater incidental detections as a result of improved awareness and advances in imaging technologies and endoscopies [3].

NENs present diverse clinical features and behaviors, making their management a multidisciplinary challenge. Their classification is based on factors such as location, morphology, stage, clinical symptomatology, markers of proliferation, and occasionally the mutation spectrum [4]. They can be categorized as functioning or non-functioning depending on hormone-induced clinical symptoms [4].

Various imaging modalities, including computed tomography (CT), magnetic resonance imaging (MRI), and positron emission tomography/computed tomography (PET/CT) are utilized for tumor detection and staging. However, none of these methods surpasses the others in detecting NENs, often necessitating a combined approach for accurate detection [5,6].

The objective of this review is to offer an overview of the latest developments in imaging and the management of these tumors while also discussing potential challenges in diagnosis and follow-up.

## 2. Country-Related Trends in Epidemiology

In the United States, it is estimated that more than 12,000 patients receive a diagnosis of neuroendocrine tumor (NET) each year, and approximately 171,000 people are presently living with this condition [3]. The rise in NET prevalence can be attributed to improvements in diagnostic techniques like advanced imaging and endoscopy along with heightened clinical awareness of these tumors [3]. This trend is not limited to the United States; NETs are on the rise globally, with variations in incidence depending on the primary site of origin. These geographical disparities are evident in the distribution of NETs across different races. Among the grade groups, the prevalence increased the most in G1 NETs, and among the sites, the prevalence was highest in the rectum, followed by the lung and small bowel. The heightened prevalence is largely linked to increased life expectancy [7].

Rectal NENs (rNENs) are more commonly found among Asians and African Americans. Moreover, in Korea and Japan, rNENs constitute as much as 60% of all GEP-NENs [8]. Conversely, small bowel NENs (SB-NENs) predominate in white Americans and Europeans [9]. Oceania demonstrates its own unique pattern, with the appendix (24.9%) being the most prevalent site, followed by the small bowel (19.5%), the lung (19.0%), and the rectum (14.6%) [10]. These remarkable disparities can be attributed to differences in classification, the compilation of databases, colonoscopy screening programs, and ethnic diversity, suggesting the potential impact of genetic and environmental factors on the etiology of NENs [9].

## 3. Pathology and Radiology: A Symbiotic Partnership

The correlation between histopathology and imaging is central to achieving a better knowledge of the imaging phenotype of NENs. The most notable change in the 2019 World Health Organization (WHO) classification is the clear separation of two distinct biologically dissimilar groups within high-grade NENs: G3 well-differentiated NETs and poorly differentiated NECs [1]. Indeed, these conditions differ significantly in terms of their origins, genetic composition, treatment strategies, and prognoses. The shared characteristic among these distinct tumors is the presence of neuroendocrine markers, encompassing synaptophysin, chromogranin A, insulinoma-associated protein 1 (INSM1), and CD56 [1].

Well-differentiated NETs (G1 and G2) are neoplasms characterized by low malignant potential and generally associated with a favorable prognosis [1]. G3 NETs display typical neuroendocrine cell morphology but with the presence of >20 mitoses per 2 mm^2^ or a Ki67 >20% [11]. Conversely, poorly differentiated NECs (including small-cell and large-cell NECs) are intrinsically high-grade tumors and are, therefore, poorly differentiated, and their tumor cells exhibit apparent malignancy without a distinct resemblance to neuroendocrine cells, even though they do express neuroendocrine differentiation markers. NECs are frequently characterized by substantial necrosis, a feature uncommon in NETs, and this can be identified through imaging techniques [1,11].

Accurate pathological diagnosis of high-grade NENs is crucial for effective clinical management, especially considering the genetic and prognostic disparities between NETs and NECs. For instance, platinum-based chemotherapy serves as a common treatment strategy for NECs, whereas NETs are typically addressed through resection often followed by other medical interventions such as capecitabine, temozolomide, everolimus, sunitinib, and somatostatin (SST) receptor-directed peptide receptor radionuclide therapy (PRRT) [4]. Notably, G3 NETs, despite their high proliferation rate, exhibit limited responsiveness to platinum-based chemotherapy.

NENs can be categorized as either functioning or non-functioning, based on their associations with specific hormone-induced clinical symptoms. Even the “non-functioning” NENs usually secrete some hormones but of an amount or type insufficient to cause a clinical syndrome.

In the case of these non-functioning NENs, the initial stages might thus not display symptoms, leading to more significant tumor growth and larger sizes upon diagnosis. These tumors often exhibit diverse enhancement patterns due to factors like necrosis and hemorrhage [12]. Conversely, functioning NENs tend to be smaller at the time of diagnosis, posing a greater challenge for detection through CT and MRI [13].

Another WHO important update includes the classification of tumors as MiNENs if they comprise a minimum of 30% of both NETs and non-NETs [1].

For Lu-NETs, the classification differs: TCs are well-differentiated, low-grade tumors with fewer than 2 mitoses per 2 mm^2^ and no necrosis. ACs are intermediate-grade tumors with 2–10 mitoses per 2 mm^2^ and/or the presence of necrosis. LCNECs and SCLCs are poorly differentiated, high-grade tumors characterized by a high mitotic index (often greater than 10 per 2 mm^2^) and extensive necrosis [14].

The expert consensus document by the European Neuroendocrine Tumor Society (ENETS) provides guidance for standardized radiological reporting templates for the diagnosis, grading, staging, and treatment of NENs. Separate, detailed templates were developed for initial tumor staging and follow-up, tailored for different types of NENs (bronchial, pancreatic, gastrointestinal) using CT and MRI, with structured sections for comprehensive reporting and assessment of the therapy response [15].

## 4. Tools and Pitfalls in Radiology and Nuclear Medicine

### 4.1. Morphologic Imaging of NENs

Morphologic imaging, encompassing CT and MRI, plays a crucial role in the detection, diagnosis, staging, and subsequent restaging of these tumors [16]. Functional characteristics, including cellular density and blood flow, can be evaluated through specific CT and MRI techniques such as perfusion and diffusion-weighted imaging (DWI) [17]. The tumor size, the number of lesions, and the enhancement patterns impact the sensitivity of multiphasic CT for detecting pancreatic NENs (pNENs) and SB-NENs (Figure 1) [18,19]. In the initial evaluation of pNENs, CT showed detection ranges from 69% to 94% [20,21]. The arterial phase is the most advantageous due to the hypervascular nature of pNENs with higher sensitivity (83–88%) compared to the portal venous phase (11–76%), particularly for smaller tumors like insulinomas [22,23]. Indeed, insulinomas show uniformly increased enhancement in the arterial phases compared to gastrinomas, which are less vascular and may exhibit more pronounced delayed enhancement due to coexistent fibrosis (Figure 2) [20]. The involvement of the main pancreatic duct in pNENs indicates higher-grade tumors, with CT reaching 88.8% sensitivity and 92.8% specificity and MRI reaching 100% sensitivity and 95.2% specificity [24,25].

However, MRI sensitivity can vary based on the tumor size, being 60–95% for lesions >2.5 cm versus 34% for lesions <1.5 cm [26]. On the other hand, CT often outperforms MRI in the assessment of vascular invasion of the tumors [25], and it is also mandatory to identify regional and distant metastases in disease staging once the primary site has been established [22]. Spectral CT, particularly with monoenergetic low keV (55 keV) images, has the potential to enhance pNEN visibility [23,27].

In SB-NENs, the sensitivity could reach 82% when it is associated with mesenteric lymphadenopathy or a desmoid reaction. For enhanced visualization of SB-NENs, employing a CT enterography protocol that includes low-density neutral oral contrast and spasmolytic agents is recommended. Notably, CT enterography has demonstrated a high detection accuracy, with a specificity of 85% and a sensitivity of 97% in identifying primary SB-NENs (Figure 3) [28,29]. In the realm of SB-NENs, the distal ileum is the most common site, often arising within the last 100 cm of the ileum [30]. For lymph node and liver metastases detection, CT has a sensitivity ranging from 60–70% and 75–100%, respectively [22].

Regarding gastric NENs, they are categorized into three distinct types. Type 1 and type 2 present as small, single or multiple polypoid lesions with strong arterial enhancement and well-defined margins. On the other hand, type 3 NENs appear as single, large, and variably enhancing masses, making it challenging to differentiate them from other more common gastric malignancies [31]. However, the identification of metastatic lymph nodes and the observation of intact overlying mucosa with mucosal tenting serve as valuable CT indicators to distinguish G1 gastric NETs from both G3 gastric NETs and gastric adenocarcinomas. Additionally, a higher metastasis-to-liver attenuation ratio can be a highly accurate tool for distinguishing hepatic metastases of gastric NETs from those originating from gastric adenocarcinomas [32].

In the case of rNENs, these tumors are usually small and localized at the time of diagnosis. Conversely, colonic NENs tend to be aggressive, poorly differentiated, large, and typically metastatic, presenting as ulcerate, infiltrative masses without intact overlying mucosa at the initial presentation [30,33]. Marked enhancement during arterial phase imaging, a larger size and a greater number of pathological lymph nodes, and a distinctive wash-in/wash-out enhancement pattern of liver metastases can serve as valuable indicators for distinguishing poorly differentiated colorectal NECs from adenocarcinomas [33].

In the case of Lu-NENs, CT also allows the diagnosis of diffuse idiopathic pulmonary neuroendocrine cell hyperplasia of the lung (DIPNECH), which exhibits multiple pulmonary nodules and bronchial wall thickening of the bronchial walls leading to small airway obstruction, mosaic attenuation during inhalation, and signs of air entrapment during expiration [34,35]. Additional CT scans during expiration are often necessary to identify air trapping, which indirectly suggests the characteristic small airway blockage associated with DIPNECH [34,35]. Lung nodules ≥5 mm in the context of DIPNECH should raise suspicions of Lu-NENs [36].

In regard to liver metastases, MRI reaches an overall sensitivity of up to 95% in their detection (Figure 4); analysis of the contribution of each specific MR technique demonstrates a sensitivity of 71.6% for DWI, 55.6% for T2-weighted (T2w) sequences, and 47.5–48.1% for dynamic contrast-enhanced T1-weighted (T1w) sequences. Moreover, the sensitivity can be further improved using hepatocyte-specific contrast agents (Figure 5) [37,38]. Hypervascular pNENs are detected using dynamic arterial phase imaging. Non-hypervascular tumors can be detected using DWI and T1w fat-saturated images. Indeed, pre-contrast T1w fat-saturated images are crucial to ensure the visibility of some tumors that might otherwise blend into the background parenchyma due to similar signal intensities on post-contrast images (Figure 6) [17]. However, post-contrast images are usually helpful for both primary and secondary detection, particularly for recognizing the enhancing rim around central cystic/necrotic change that may otherwise be mistaken for a smaller pancreatic cyst on T2w images.

Besides identifying liver metastases, DWI can also be utilized to assess treatment response [38,39].

MRI, boasting superior soft tissue contrast, outperforms CT in rendering detailed images of the liver and pancreas. However, its usage as a whole-body screening methodology is hampered by several challenges and, therefore, MRI is more commonly employed to assess single organs or single anatomic areas, such as the upper abdomen. Therefore, MRI is commonly reserved for cases of high clinical suspicion of specific NEN types in single organs or single anatomic areas, such as pNENs and rNENs. It is also used as a diagnostic aid when CT scans yield inconclusive results or when magnetic resonance cholangiopancreatography (MRCP) is required to evaluate the biliary tract (Figure 7) [16].

MRI stands out as the most sensitive technique for evaluating bone metastases, especially early osseous metastases, which might be overlooked by CT. In this context, bone abnormalities appear as areas of hyperintensity on T2w and on short tau inversion recovery (STIR) sequences and strongly enhance after gadolinium injection [40,41,42].

Despite the continuing progress in imaging performance, determining the tumor grade non-invasively remains a challenge. On MRI, higher-grade tumors in the NET category show greater diffusion restrictions in DWI and intra-voxel incoherent motion DWI. However, the apparent diffusion coefficient (ADC) provides inconsistent values among tumors of the same grade, compounded by biological heterogeneity and technical factors influencing the reproducibility of ADC measurements, making the use of this technique still not ready for routine clinical practice [43,44,45].

### 4.2. Hybrid Imaging of NENs

Metabolic imaging, specifically PET/CT utilizing radiolabeled somatostatin analogues (SSAs) like DOTA, which displays an especially high affinity toward SST receptor 2, is strongly recommended for tumor staging, preoperative evaluation, and resection planning of NENs (Figure 8). Gallium-68 (68Ga) or Copper-64 (Cu64)-DOTA-peptides PET/CT, notably [68Ga]Ga-DOTATOC, [68Ga]Ga-DOTANOC, and [68Ga]Ga-DOTATATE, and [64Cu]Ga-DOTATATE represent the gold standard in functional imaging radiopharmaceuticals. Moreover, DOTATATE can be radiolabeled with lutetium-177 (177Lu) to create [177Lu]-DOTATATE for PRRT [46].

DOTA PET/CT substantially improves the sensitivity and specificity of NEN detection, achieving 80% to 98% detection rates [5,47], even in cases of lesions affecting the lymph nodes, bones, liver, and peritoneum (Figure 9). Conversely, when assessing poorly differentiated NENs or tumors with high proliferative activity, PET/CT with 18-fluoro-2-deoxyglucose (18F-FDG) is the favored option [13,46,48].

In general, [68Ga]DOTATATE PET/CT exhibits superior diagnostic performance when compared to [18F]F-FDG PET/CT, which typically shows a lower sensitivity in the range of 50% to 70% (Figure 10) [40,49].

However, the possibility of high DOTA uptake in normal tissues, particularly in the uncinate process and pancreatic head region, given the high concentration of SST receptor 2 positive endocrine islets in those areas, can make [68Ga]DOTATATE PET/CT scans challenging to be interpreted. Therefore, it is essential to establish a correlation with morphologic imaging (CT or MRI) to exclude any pathological findings in these areas [50]. False positive DOTA results may also occur due to non-malignant conditions and inflammatory/infectious states, as SST receptor expression can be elevated in activated lymphocytes. Conversely, false negatives might result from factors like lesion dedifferentiation, small lesion size, and histotypes with low SST receptor expression [47,51,52].

Additionally, the accumulation of [68Ga]Ga-DOTA in the spleen makes this tracer unsuitable for differentiating intrapancreatic splenules from pNENs [46].

Despite its well-recognized contribution to NEN detection and staging, [68Ga]DOTATATE PET/CT lacks well-defined response criteria and is not commonly employed for interim or end-of-treatment assessment. For example, to monitor PRRT, a combination of [68Ga]DOTATATE PET/CT and triple-phase contrast-enhanced CT and/or MRI is frequently preferred [53]. High radiotracer cost is also a barrier to [68Ga]DOTATATE PET/CT clinical use. A recent study [54] introduced a three-point scale for evaluating the concordance or discordance between 68Ga-DOTA-peptides/FDG PET/CT findings. This scale categorizes patients as patients with [18F]F-FDG-negative findings; patients with [18F]F-FDG-positive findings where all lesions detected by [18F]F-FDG-PET/CT also show positivity in [68Ga]DOTATATE PET/CT; and patients with [18F]F-FDG-positive findings where there are multiple [18F]F-FDG-positive lesions but at least one of them is negative on [68Ga]DOTATATE PET/CT. This functional classification system was more effective than traditional pathological grading in predicting patient outcomes. It allowed for the classification of patients into three distinct survival groups, each with statistically significant differences in the median progression-free survival.

Notably, higher values of uptake on [18F]F-FDG PET/CT are employed to assess their potential correlation with lower response rates to PRRT [55,56]. Regarding the correlation between the tumor grade and SST receptor expression utilizing 68Ga-DOTATOC’s mean standardized uptake value (SUVmean) in NENs, the initial findings indicate a heightened SUVmean in G2 NETs in contrast to G1 NETs. Moreover, a hybrid positron emission tomography/magnetic resonance imaging (PET/MRI) biomarker, the SUVmean/ADC ratio, has proven to be a reliable indicator of the tumor grade, with high sensitivity (100%) and specificity (86%) [57]. Additionally, different baseline maximum standardized uptake value (SUVmax) values have been proposed to predict PRRT response, for example, SUVmax > 17.0, 100% sensitivity and 85% specificity; SUVmax > 16.4, 95% sensitivity and 60% specificity; and SUVmax > 13.0, 83% sensitivity and 80% specificity [58,59,60].

[18F]F-FDG PET/CT provides vital diagnostic information but faces difficulties in precisely identifying NENs. [18F]F-FDG uptake indicates dedifferentiation, increases with higher tumor grades, and correlates with poorer survival and reduced response to SST receptor PRRT treatment. However, [18F]F-FDG PET/CT cannot replace traditional tumor grading due to complex factors affecting [18F]F-FDG uptake. Survival outcomes are linked to the balance between the SST receptor and [18F]F-FDG uptake, formalized in the NETPET score [61].

The NETPET score is a system used to evaluate and grade NETs combining findings at [68Ga]Ga-DOTA-peptides and [18F]F-FDG PET/CT into a single parameter. This score typically ranges from P0 to P5. P1 indicates purely somatostatin receptor-positive lesions without significant [18F]F-FDG uptake. P5 indicates the presence of significant [18F]F-FDG-positive/somatostatin receptor-negative disease. P2 to P4 represent intermediate categories where the “target” lesion shows increasing [18F]F-FDG uptake relative to somatstatin receptor imaging from P2 to P4. P0 indicates a normal scan on both [18F]F-FDG and somatostatin receptor imaging, potentially in cases of completely resected disease [62].

The role of hybrid and/or functional image guidance becomes crucial in the selection of appropriate targets for tissue biopsies. Morphological imaging using methods such as tumor enhancement and DWI restriction contributes to identifying viable tumor tissue for sampling [63]. For metabolic imaging, [18F]F-FDG PET/CT aids in identifying metabolically active and aggressive lesions. However, it is essential to acknowledge the increasing importance of intra- and interlesional heterogeneity guided by tumor genomics in treatment decisions [64].

The emerging radiopharmaceuticals fluorine-labeled SST receptor-agonists ([18F]F-AlF-NOTA-Octreotide and [18F]F-SiFAlin-TATE) and [64Cu]Cu-SARTATE offer practical advantages over existing options [65,66]. SST receptor antagonists like [68Ga]Ga-OPS202 show favorable biodistribution and higher detection rates, particularly at the liver level [67]. [68Ga]Ga-NODAGA-LM3 and [68Ga]Ga-DOTA-LM3 are promising SST receptor 2-antagonists for NETs [68]. [68Ga]Ga-Exendin is employed for detecting insulinomas, which are DOTATATE negative in up to 50% of the cases given low SST receptor 2 expression, and [18F]F-meta-fluorobenzylguanidine ([18F]F-MFBG) shows encouraging results for neuroblastoma imaging [69].

PET/MRI holds several advantages over PET/CT for imaging NETs, mainly due to MRI’s superior soft tissue contrast resolution and synchronous acquisition of the PET and MRI data. Comparative studies of PET/MRI and PET/CT have shown superior PET/MRI efficacy in the detection of bone and liver metastases.

Moreover, the simultaneous acquisition of PET and MRI in PET/MRI enables superior image registration, motion-correction approaches, and the generation of higher-quality fused images [70,71]. DOTA-PET/MRI outperformed DOTA-PET/CT in detecting malignant NENs: accuracy (97% vs. 94.6%), detection rates (72.5% vs. 62.7%), and correctly classified more lesions than PET/CT (90.8% vs. 86.7%). Moreover, PET/MRI, through ADC, provided information regarding the treatment response and final outcomes [72,73,74,75].

## 5. Spectrum of Therapeutic Options

For G1-G2 NETs, surgery is the primary treatment for resectable GEP-NETs, including those with liver metastases, as it improves symptom control and overall survival if safely resected and/or ablated [76].

Long-acting SSAs like octreotide and lanreotide are used for symptomatic or progressive disease, enhancing symptom management and progression-free survival. PRRT is effective for inoperable/metastatic G1/G2 GEP-NETs that progress despite SSA therapy. Active surveillance is suitable for asymptomatic patients with low tumor volume or high surgical risk. G3 NETs pose distinct challenges, and surgical resection stands as the primary approach for eligible patients. In general, the surgical approach is strictly dependent on the primary organ of origin [76]. The utilization of adjuvant chemotherapy lacks substantial supporting evidence in the context of fully resected GEP-NETs [76].

For unresectable tumors, the first-line therapy is based on long-acting SSAs, like octreotide and lanreotide, which are used for both functioning and non-functioning NETs due to their antiproliferative activity, as demonstrated in the PROMID and CLARINET studies, enhancing symptom management and progression-free survival [77,78,79]. However, SSAs may disrupt gallbladder motility by affecting enzyme cascades, leading to the formation of gallstones. Therefore, it is recommended to consider prophylactic cholecystectomy during the initial surgery for NETs to prevent future emergency surgeries for gallstone complications [80]. Systemic chemotherapy can be recommended for symptomatic, higher-volume disease or patients with progression after initial surveillance.

The treatment of G3 NETs involves multiple options, including systemic chemotherapy, PRRT, and targeted therapies. Systemic chemotherapy remains a cornerstone for high-grade tumors, particularly those with high proliferative rates; it is advisable to use a neoadjuvant approach with caution as the effects on quality of life and long-term results in terms of prolonged survival remain yet to be confirmed [81].

PRRT can be used as an initial treatment for G3 NETs with Ki-67 ≤ 55% and high SST receptor expression over platinum-based chemotherapy due to its use of radiation-labeled synthetic SSAs for more effective therapy. This therapy is effective because it specifically targets cancer cells that have an abundance of SST receptors. When the radiolabeled drug binds to these receptors on cancer cells, it prompts internalization, thereby enhancing the therapeutic impact of the treatment. Accordingly, following the North American Neuro-Endocrine Tumor Society (NANETS) guidelines, NENs with high expression of SST receptors are considered prime candidates for PRRT [82,83]. PRRT, on the other hand, is approved for inoperable/metastatic G1-G2 GEP-NETs that have progressed despite therapy with SSA [83].

In phase III trials, molecularly targeted therapies such as everolimus and sunitinib enhanced the progression-free survival but lacked significant tumor response induction [84].

Additionally, everolimus helps protect against renal failure caused by other drugs, which is particularly important for transplant patients who are at a higher risk of developing cancer [85].

In the case of well-differentiated NETs or larger disease volumes, the CAPTEM regimen or carboplatin (or cisplatin) combined with etoposide is considered a viable option for cytotoxic chemotherapy [86]. Active surveillance is a suitable approach for patients with asymptomatic GEP-NETs with low tumor volume or those with a resectable disease but facing a high surgical risk [76,87].

Intra-arterial therapies targeting metastatic liver disease are guided by imaging characteristics to identify appropriate candidates. These strategies encompass a range of approaches, including radiofrequency ablation (RFA), microwave ablation (MWA), and selective internal radiation therapy (SIRT) [88]. Percutaneous RFA can be performed if the target metastases are visible on ultrasound or CT without contrast injection, but it becomes more difficult if the metastasis is visible only on MRI or on arterial phase contrast-enhanced CT [88].

Even for Lu-NENs, the primary treatment approach is surgical resection, particularly for typical lung carcinoids, which generally results in an excellent prognosis [89]. Adjuvant therapy is not recommended for patients with TCs or stage II or lower ACs. However, the five-year survival for ACs (intermediate grade) is less favorable. Some experts suggest considering adjuvant cisplatin and etoposide, possibly with radiation therapy, for patients with resected stage IIIA ACs. Nevertheless, the use of adjuvant therapy in this context remains a subject of debate [90].

## 6. Follow-Up Imaging: Strengths and Challenges

Imaging modalities have varying strengths and limitations, which can result in false positive or negative findings [91].

[68Ga]DOTATATE PET/CT is a highly effective imaging technique for diagnosing and locating primary tumors in patients with metastatic NENs, whether of known or unknown origin. It plays a fundamental role in both the initial diagnosis and ongoing follow-up [46,60,92].

After undergoing locoregional treatments, with a primary focus on liver metastases, the application of modified response evaluation criteria in solid tumors (mRECIST) involves employing CT or MRI with arterial phase acquisition to identify tumor necrosis [93,94]. The criteria for evaluating the response encompass a reduction of 25% or greater in arterial enhancement or 50% or greater in venous enhancement, pointing to a more positive prognosis [95].

However, limitations in both RECIST and RECIST 1.1 for defining the progression of metastases in NENs treated with antiangiogenic drugs or loco-regionally have been identified [96]. Specifically, due to the slow growth of most well-differentiated NETs, it may take a considerable amount of time to detect an increase in size of ≥20%, a parameter required by RECIST for defining progression. Furthermore, RECIST criteria do not assess osseous metastases unless they possess a measurable soft-tissue component, and in numerous cancer types, including NENs, metastatic disease can exist within lymph nodes that do not meet RECIST criteria for enlargement [97]. As a result, RECIST criteria may underestimate the treatment efficacy and detect tumor growth at a later stage [98].

Enhanced tumor vascularization on CT and MRI allows for the efficient delivery of therapeutic agents via intra-arterial treatments [88]. However, it is also a negative prognostic factor, suggesting a more aggressive tumor phenotype. Follow-up imaging is typically scheduled 2–4 weeks after intra-arterial therapy to gauge its effectiveness and determine the need for subsequent treatments [93,99].

RFA aims to ablate not only the tumor tissue but also a margin of the surrounding healthy liver tissue to achieve “margins of safety,” which can lead to a larger area of abnormality surrounding the ablated tumor, making interpreting MRI and CT challenging in the early stages of follow-up due to the associated intense inflammatory response [91]. Consequently, during the initial stages of follow-up after RFA, the interpretation of CT and MRI results can be challenging due to this robust inflammatory response associated with the expanded abnormal tissue surrounding the treated tumor [91].

Determining the ideal imaging technique and timing for evaluating the response to SIRT remains a subject of ongoing discussion, despite the presence of diverse imaging findings linked to the procedure itself. Nevertheless, early follow-up imaging should be cautiously used to avoid any misinterpretation of ablation-triggered inflammation as a residual or progressed disease [92]. Indeed, signs of a tumor response become discernible later within 3 to 6 months after the procedure, emphasizing the need for a thorough and patient-centered assessment [92].

During post-PRRT follow-up, preventing misinterpretations is vital. Early after treatment, it is important not to mistake persistent arterial phase enhancement, caused by parenchymal inflammation, for disease progression [93]. Pseudoprogression, a radiation-triggered inflammatory reaction leading to an apparent increase in imaging size during and after therapy, is a frequent occurrence in NENs [94]. While it is recommended to wait for 2–3 months post PRRT completion before reassessing the tumor, definitive guidelines for this timeline are not established. For patients with a low-grade, localized-stage disease or those with indolent tumors, the intervals between follow-ups should be longer (e.g., every 9–12 months) (Figure 11) [95].

Conversely, individuals bearing poor prognostic indicators like high-grade, extensively metastatic [18F]F-FDG-avid disease, aggressive tumor behavior, or severe endocrinopathy necessitate more frequent imaging. This should be performed by utilizing suitable conventional and molecular imaging techniques tailored to the specific case [95].

[18F]F-FDG PET/CT has been found to detect a response more quickly and accurately than CT in hypovascular liver metastases after SIRT. PET is also useful in distinguishing viable neoplastic tissue from edema, hemorrhage, or fibrosis after treatment, while CT is better equipped to identify diffuse parenchymal changes and complications such as cholecystitis or bilomas [96].

In patients treated using SSA/PRRT, to prevent any potential impact on radiotracer biodistribution that could affect the interpretation of [68Ga]DOTATATE PET/CT, SSA therapy should be continued as maintenance therapy after PRRT. Indeed, long-acting SSA therapy usually increases the degree of uptake of [68Ga]DOTA within metastases and, consequently, increased uptake alone may not accurately represent true progression [97,98]. The European Association of Nuclear Medicine procedure recommend waiting for 3 to 4 weeks after long-acting analogue administration before conducting [68Ga]DOTATATE PET/CT [99].

Regarding the follow-up post NEN surgical resection, the NANETS guidelines do not recommend the use of SST receptor PET scans. Instead, it advises using CT scans for regular follow-up and reserving the use of SST receptor PET scans for patients exhibiting clinical concerns regarding disease progression that may not be apparent on cross-sectional imaging (Figure 12) [83,95]. Hence, the determination of response in NENs is relative and ultimately depends on the specific treatment or treatments provided to the patient, tumor kinetics or grade, and the anticipated outcome [95].

## 7. Future Frontiers in Radiomics

Radiomics is a promising field in NEN management, offering advanced imaging analysis for enhanced tumor characterization, prognostication, and treatment response assessment. The fusion model combining radiomics signatures and radiological characteristics showed good performance in predicting the grade of nonfunctioning pancreatic neuroendocrine tumors, with AUCs of 0.956 in the training set and 0.864 in the testing set in the paper of Zhu et al., 2024 [100]. Ye et al. showed an interpretable radiomics-based random forest model that can effectively differentiate between G1 and G2/3 pancreatic NETs, demonstrating favorable interpretability [101]. Advancements in the radiomics field can significantly improve personalized treatment strategies for NENs in the future.

## 8. Conclusions

In conclusion, the integration of diverse morphologic and hybrid imaging techniques such as CT, MRI and PET/CT has significantly advanced the detection and characterization of NENs across various organs. Imaging findings guide treatment decisions for NENs, assisting in the choice between surgery, medical therapies, and interventional approaches. They also aid in identifying candidates for targeted treatments like PRRT and intra-arterial therapies for liver metastases. However, challenges persist in non-invasive tumor grading and distinguishing between tumor types. Nevertheless, the combination of imaging with genetic and molecular profiling improves treatment decisions. Looking ahead, emerging radiopharmaceutical peptides for PET/CT and PET/MRI hold promise, with the latter offering superior soft tissue contrast resolution for more precise lesion localization.

## Figures and Tables

**Figure 1 jcm-13-03281-f001:**
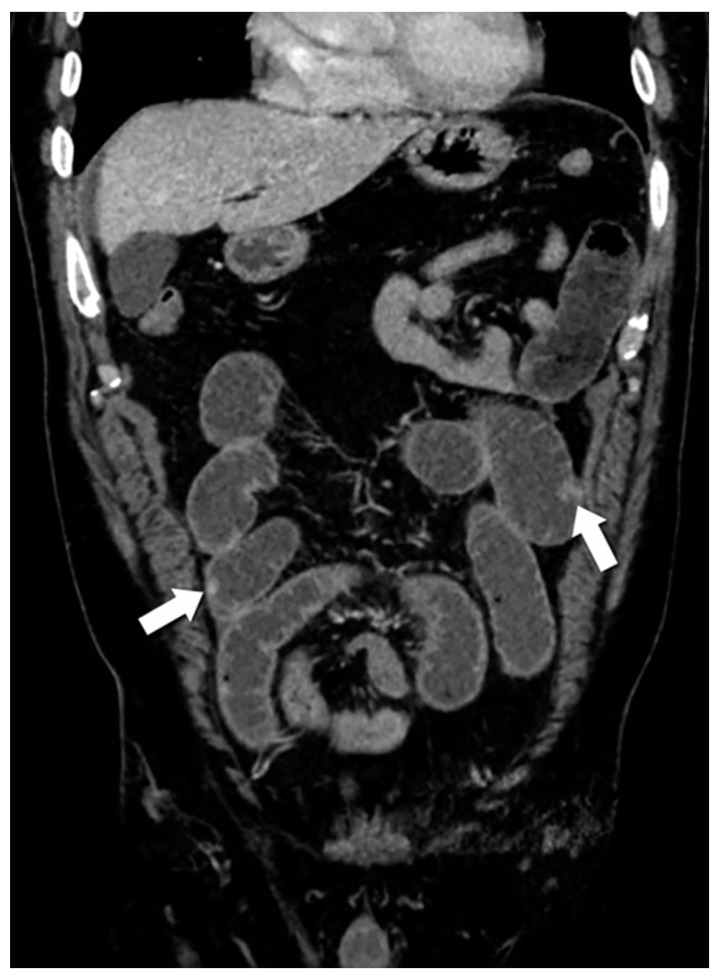
Multiple small bowel NENs. Coronal contrast-enhanced (CE)-CT demonstrates multiple small NENs (arrows) scattered throughout the small bowel. The presence of multiple lesions may discourage the use of a laparoscopic approach and instead suggests laparoscopic palpation and resection or laparotomy.

**Figure 2 jcm-13-03281-f002:**
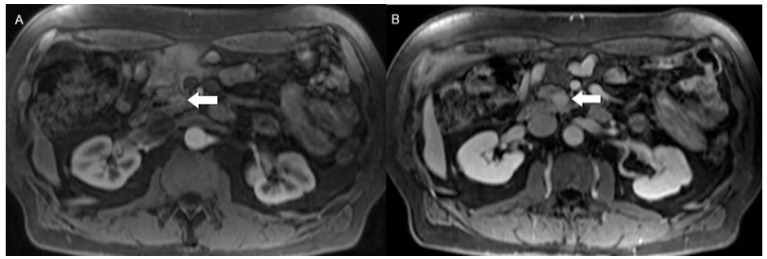
Uncinate process gastrinoma. (**A**) Arterial phase axial CE-MRI reveals subtle and indistinct enhancement of a solid lesion in the uncinate process (arrow). The degree of enhancement of the lesion increases at (**B**) delayed phase axial CE-MRI. Pathology confirmed the diagnosis of gastrinoma; these tumors are typically less vascularized and more fibrotic than other NEN subtypes, accounting for a more pronounced delayed enhancement.

**Figure 3 jcm-13-03281-f003:**
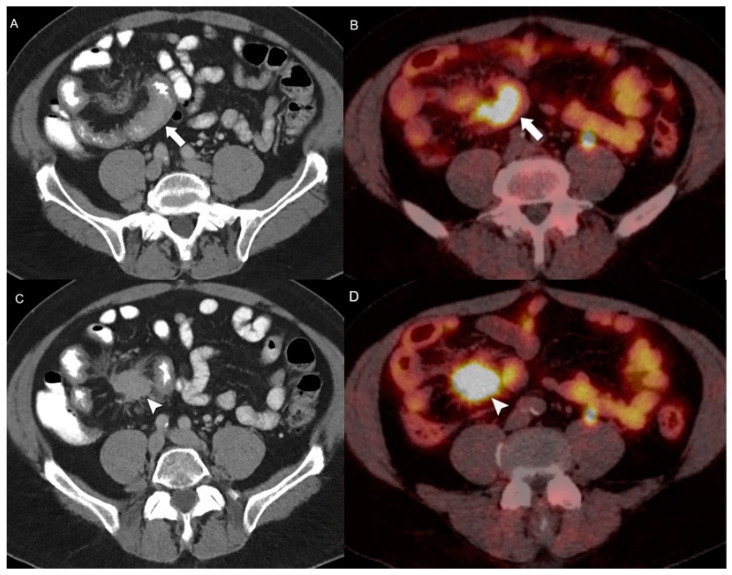
Small bowel NEN with a mesenteric desmoid reaction on entero-PET/CT. (**A**) Axial CE-CT enterography and (**B**) [68Ga]DOTATATE PET/CT show focal small bowel wall thickening with associated intense uptake confirming the neuroendocrine nature of the lesion (arrow). Associated mesenteric neuroendocrine mass (arrowhead) on axial CE-CT (**C**) and [68Ga]DOTATATE PET/CT (**D**) markedly uptake (arrowhead) DOTATATE, infiltrates the mesentery and tethers the small bowel.

**Figure 4 jcm-13-03281-f004:**
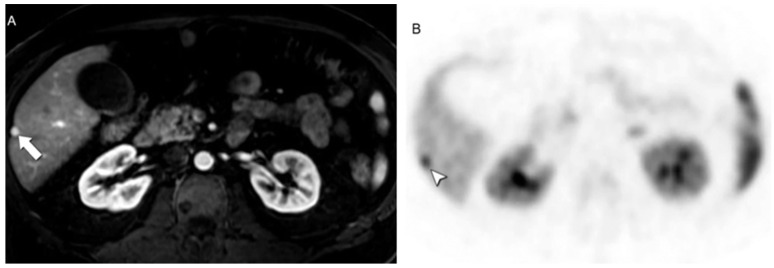
Classic presentation of liver metastases from NEN. (**A**) Axial CE-MRI obtained during the arterial phase reveals a markedly enhancing lesion in hepatic segment VI (arrow). (**B**) On [68Ga]DOTATATE PET/CT the lesion demonstrates avid radiopharmaceutical uptake (arrowhead).

**Figure 5 jcm-13-03281-f005:**
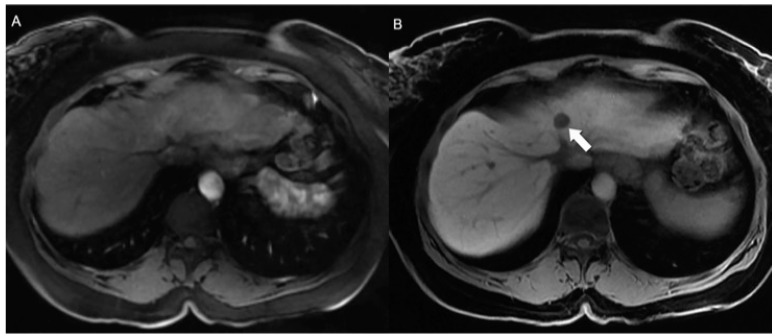
Hepatocyte-specific contrast agents can improve NEN liver metastasis detection. (**A**) A metastasis in segment IV only mildly enhances during arterial phase imaging, making its detection challenging. (**B**) NEN metastases to the liver, lacking hepatocyte and bile ductules, cannot concentrate hepatospecific contrast media, and, therefore, they stand out as hypointense lesions against enhanced normal adjacent hepatic parenchyma in the hepatospecific phase (arrow).

**Figure 6 jcm-13-03281-f006:**
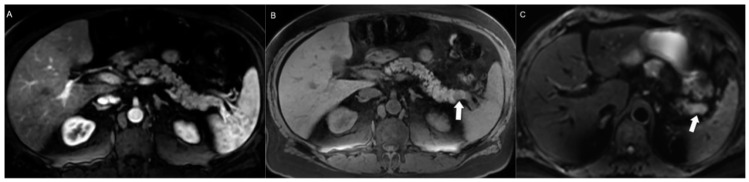
Contribution of T1w non-contrast enhanced images to detecting pancreatic NENs. (**A**) A pancreatic tail NEN could be missed on this axial arterial phase CE-MRI (**A**) given its subtle and poorly defined enhancement. (**B**) Non-contrast axial T1-weighted fat-saturated image helps in detecting the tumor (arrow) given its low signal intensity compared to the surrounding T1w hyperintense parenchyma. (**C**) Additionally, diffusion-weighted imaging (DWI) images can boost confidence by revealing the diffusion-restricted mass.

**Figure 7 jcm-13-03281-f007:**
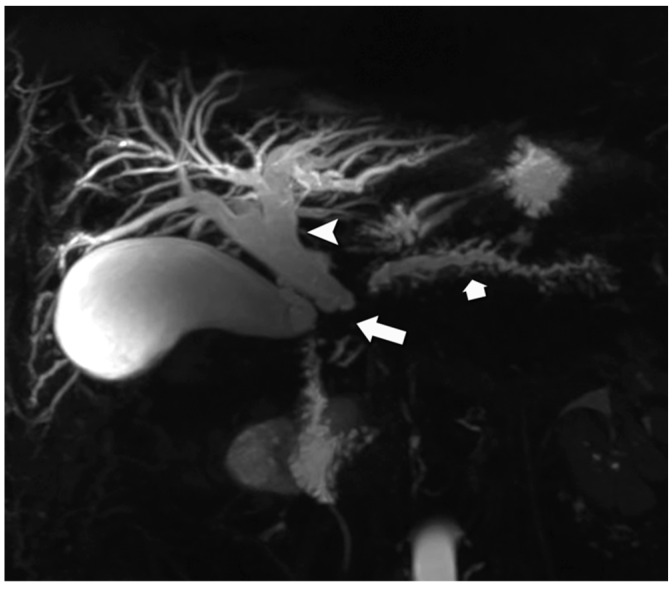
Double-duct sign in NEN. Magnetic resonance cholangiopancreatography (MRCP) image reveals an obstructing lesion at the pancreatic head region (arrow), resulting in dilation of both the biliary system (arrowhead) and the main pancreatic duct (short arrow). This sign is more typical of pancreatic ductal adenocarcinoma; however, subsequent histological examination revealed the diagnosis of a pancreatic NEN.

**Figure 8 jcm-13-03281-f008:**
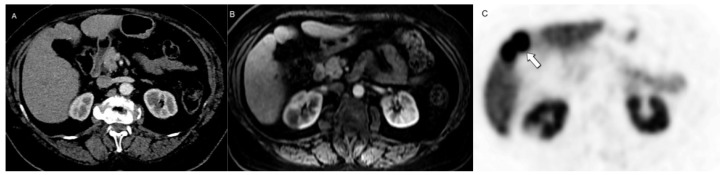
NEN liver metastases. Arterial phase imaging as by (**A**) CE-CT and (**B**) CE-MRI might miss NEN metastatic to the liver, which, in contradistinction, can be easily appreciated through (**C**) [68Ga]DOTATATE PET/CT (arrow).

**Figure 9 jcm-13-03281-f009:**
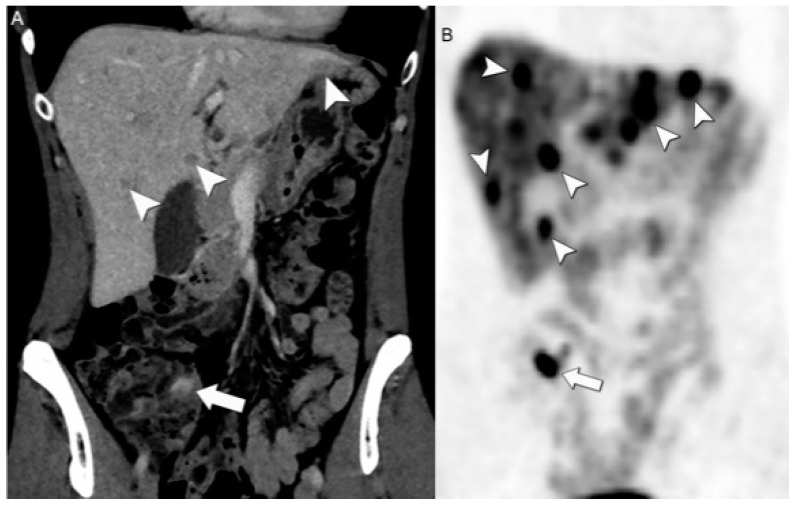
Appendiceal NEN with liver metastases. (**A**) Coronal CE-CT showed focal enhancement in the appendix (arrow) and mild peripherally enhancing hepatic lesions (arrowheads). (**B**) The marked uptake on [68Ga]DOTATATE PET/CT clearly delineates the appendiceal NEN and improves the recognition of the associated liver metastases.

**Figure 10 jcm-13-03281-f010:**
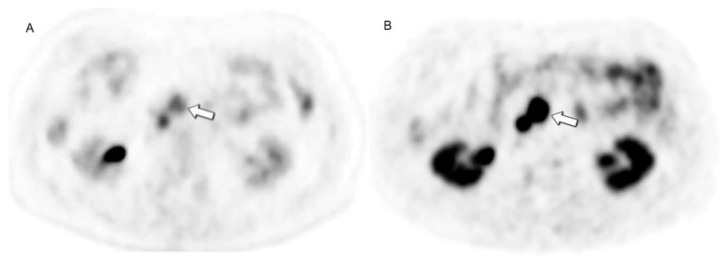
High diagnostic accuracy of [68Ga]DOTATATE PET/CT. Same patient as in Figure 2, uncinate process NEN demonstrates mild to low uptake at (**A**) [18F]F-FDG PET/CT (arrow) but intense uptake at (**B**) [68Ga]DOTATATE PET/CT.

**Figure 11 jcm-13-03281-f011:**
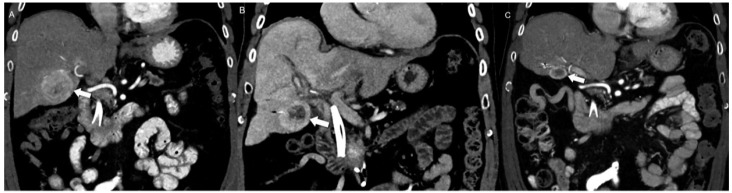
Challenges in assessing PRRT therapy. (**A**) Pre-treatment coronal CE-CT shows a markedly enhancing liver metastasis (arrow). (**B**) Follow-up coronal CE-CT performed 4 months after PRRT demonstrates unchanged size but new internal necrosis suggestive of treatment response. (**C**) 10-month follow-up coronal CE-CT shows both lesion shrinkage and further decreased enhancement, which are expected indicators of a positive response to PRRT treatment. This example highlights the complexity of evaluating treatment outcomes during the follow-up period after PRRT.

**Figure 12 jcm-13-03281-f012:**
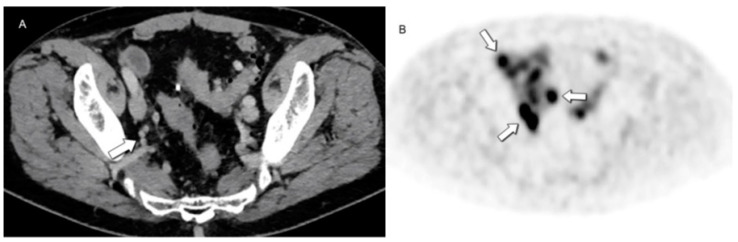
Value of metabolic imaging in detecting recurrent disease. No concerning lesions were identified at (**A**) CE-CT in this patient with increased chromogranin levels following systemic treatment. (**B**) [68Ga]DOTATATE PET/CT revealed marked uptake in multiple subcentimeter pelvic lymph nodes (arrows, retrospectively visible also in (**A**)), highlighting areas of recurrent disease that may be overlooked by morphologic imaging due to small lesion size or unexpected locations.

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
