# Peer review of "Advancements in Neuroendocrine Neoplasms: Imaging and Future Frontiers"

_jcm, 2024, doi:10.3390/jcm13113281_

Round 1

Reviewer 1 Report

Comments and Suggestions for Authors

Dear Authors,

It was a pleasure to review your work. The manuscript is a comprehensive summary of the available NEN diagnostics and possible treatment methods. I did not find any issues requiring correction. The text is well-written, without unnecessary redundancies, and allows the reader to familiarize with the topic. It also includes data on future diagnostic possibilities and the figures have accurate captions and are relevant to the text. 

Author Response

Dear Reviewer,
thank you for taking the time to review our manuscript and for your positive feedback. We are pleased that the manuscript met your expectations and grateful for your support.

Reviewer 2 Report

Comments and Suggestions for Authors

This review provides an overview of the latest developments in imaging of neuroendocrine neoplasms. It is a well-written review with a praiseworthy focus on radiology and nuclear medicine, providing interesting and informative figures. I suggest removing “management” from the title as it may be misleading and does not emphasize the main topic which is imaging. Indeed, the management usually refers to therapies that are only briefly summarized in the manuscript.

There are some minor critical issues to be resolved:

In "Pathology and Radiology: A Symbiotic Partnership," the classification according to grade (G1-G2-G3 NET) is valid for GEP-NETs, while Lu-NETs have a different histologic classification based on the mitotic index and the presence of necrosis, please detail.

I suggest citing the recent ENETS standardized (synoptic) reporting for radiological and molecular imaging in neuroendocrine tumours to enrich discussion.

The section "Spectrum of Therapeutic Options" could benefit from a reorganization, describing different therapied according to grade of differentiation, starting from grade 1-2 instead of grade 3. Although the therapy is correctly reported to depend on grade, extent of disease, and clinical condition of the patient, chemotherapy generally is not the first line therapy. The first line is based on somatostatin analogues (SSAs), which represent the cornerstone of therapy in these neoplasms. Furthermore, it seems that the role of SSAs is limited to functioning NETs, whereas, as indicated by ESMO guidelines, the PROMID study, the CLARINET study, and subsequent research, SSAs are also prescribed in non-functioning NETs due to their antiproliferative activity. I recommend citing therapeutic guidelines and the paper by Oberg K, et al. 2009 (Oberg K, Ferone D, Kaltsas G, Knigge UP, Taal B, Plöckinger U; Mallorca Consensus Conference participants; European Neuroendocrine Tumor Society. ENETS Consensus Guidelines for the Standards of Care in Neuroendocrine Tumors: biotherapy. Neuroendocrinology. 2009;90(2):209-13. doi: 10.1159/000183751. PMID: 19077379.) summarizing recent evidence about the topic. It is stated that PRRT is the preferred initial treatment for G3 NETs with Ki-67 ≤ 55%, but beyond approval limitations that should be considered, the therapeutic algorithms include other options that should be cited. With regard to PRRT I suggest reporting its neoadjuvant and upfront use by citing the following paper by Lania et al. (doi: 10.3389/fendo.2021.651438).

With regards to “future frontiers” some data about radiomics should be usefully included due to its promising role in this field.

Other minor revisions include the following:

-          Some acronyms should be revised. For example, in line 42-43, "small cell neuroendocrine carcinomas (NECs)" could be replaced with "small cell neuroendocrine carcinomas (SCLC)" and "large cell NECs" with "large cell NECs (LCLC)"; contrast-enhanced CT (line 142).

-          Lines 406-407, please rephrase.

-          Lines 344-346-347, correct "FDG" to "[18F]F-FDG".

Author Response

#Re2.1: This review provides an overview of the latest developments in imaging of neuroendocrine neoplasms. It is a well-written review with a praiseworthy focus on radiology and nuclear medicine, providing interesting and informative figures. I suggest removing “management” from the title as it may be misleading and does not emphasize the main topic which is imaging. Indeed, the management usually refers to therapies that are only briefly summarized in the manuscript.

#Re2.1: Thanks for the suggestion. We removed “management” from the title accordingly.

#Re2.2: In "Pathology and Radiology: A Symbiotic Partnership," the classification according to grade (G1-G2-G3 NET) is valid for GEP-NETs, while Lu-NETs have a different histologic classification based on the mitotic index and the presence of necrosis, please detail.

#Re2.2: Thank you for noticing this, we added a brief paragraph regarding this aspect, as follows: “For Lu-NETs, the classification differs: Typical carcinoids (TCs) are well-differentiated, low-grade tumors with fewer than 2 mitoses per 2 mm² and no necrosis. Atypical carcinoids (ACs) are intermediate-grade tumors with 2-10 mitoses per 2 mm² and/or the presence of necrosis. Large cell neuroendocrine carcinomas (LCNECs) and small cell lung carcinomas (SCLCs) are poorly differentiated, high-grade tumors characterized by a high mitotic index (often greater than 10 per 2 mm²) and extensive necrosis”.

#Re2.3: I suggest citing the recent ENETS standardized (synoptic) reporting for radiological and molecular imaging in neuroendocrine tumours to enrich discussion.

#Re2.3: Thanks for this suggestion, we added the references for ENETS standardized reporting, with a paragraph as follows: “The expert consensus document by the European Neuroendocrine Tumor Society (ENETS) provides guidance for standardized radiological reporting templates for the diagnosis, grading, staging, and treatment of NENs. Separate, detailed templates were developed for initial tumor staging and follow-up, tailored for different types of NENs (bronchial, pancreatic, gastrointestinal) using CT and MRI, with structured sections for comprehensive reporting and assessment of therapy response”.

#Re2.4:The section "Spectrum of Therapeutic Options" could benefit from a reorganization, describing different therapied according to grade of differentiation, starting from grade 1-2 instead of grade 3.

#Re2.4:  Thanks for noticing this, we added a brief paragraph regarding G1-G2 NETs before G3 NETs; as follows: For G1-G2 NETs surgery is the primary treatment for resectable GEP-NETs, including those with liver metastases, as it improves symptom control and overall survival if safely resected and/or ablated [76].

Long-acting SSAs like octreotide and lanreotide are used for symptomatic or progressive disease, enhancing symptom management and progression-free survival. PRRT is effective for inoperable/metastatic G1-G2 GEP-NETs that progress despite SSA therapy”

#Re2.5: Although the therapy is correctly reported to depend on grade, extent of disease, and clinical condition of the patient, chemotherapy generally is not the first line therapy. The first line is based on somatostatin analogues (SSAs), which represent the cornerstone of therapy in these neoplasms. Furthermore, it seems that the role of SSAs is limited to functioning NETs, whereas, as indicated by ESMO guidelines, the PROMID study, the CLARINET study, and subsequent research, SSAs are also prescribed in non-functioning NETs due to their antiproliferative activity. I recommend citing therapeutic guidelines and the paper by Oberg K, et al. 2009 (Oberg K, Ferone D, Kaltsas G, Knigge UP, Taal B, Plöckinger U; Mallorca Consensus Conference participants; European Neuroendocrine Tumor Society. ENETS Consensus Guidelines for the Standards of Care in Neuroendocrine Tumors: biotherapy. Neuroendocrinology. 2009;90(2):209-13. doi: 10.1159/000183751. PMID: 19077379.) summarizing recent evidence about the topic.

#Re2.5: Thank you for noticing this issue: we have edited the paragraph to better clarify this, as follows: For unresectable tumors, the first line therapy is based on long-acting SSAs, like octreotide and lanreotide, which are used for are used for both functioning and non-functioning NETs due to their antiproliferative activity, as demonstrated in the PROMID and CLARINET study, enhancing symptom management and progression-free survival”. And

“Systemic chemotherapy can be recommended for symptomatic, higher-volume disease or patients with progression after initial surveillance”.

#Re2.6: It is stated that PRRT is the preferred initial treatment for G3 NETs with Ki-67 ≤ 55%, but beyond approval limitations that should be considered, the therapeutic algorithms include other options that should be cited. With regard to PRRT I suggest reporting its neoadjuvant and upfront use by citing the following paper by Lania et al. (doi: 10.3389/fendo.2021.651438).

#Re2.6:Thank you for raising this issue, we have rephrased the paragraph as follows: The treatment of G3 NETs involves multiple options, including systemic chemotherapy, PRRT, and targeted therapies. Systemic chemotherapy remains a cornerstone for high-grade tumors, particularly those with high proliferative rates; it is advisable to use a neoadjuvant approach with caution, as the effects on quality of life and long-term results in terms of prolonged survival remain yet to be confirmed”.

#Re2.7: With regards to “future frontiers” some data about radiomics should be usefully included due to its promising role in this field.

#Re2.7: Thanks for noticing this, we have added a brief paragraph on radiomics, as follows: “Radiomics is a promising field in NEN management, offering advanced imaging analysis for enhanced tumor characterization, prognostication, and treatment response assessment. The fusion model combining radiomics signatures and radiological characteristics showed good performance in predicting the grade of nonfunctioning pancreatic neuroendocrine tumors, with AUCs of 0.956 in the training set and 0.864 in the testing set in the paper of Zhu et al., 2024. Ye et al., showed an interpretable radiomics-based random forest model that can effectively differentiate between G1 and G2/3 pancreatic NETs, demonstrating favorable interpretability. Advancements in the radiomics field can significantly improve personalized treatment strategies for NENs in the future.”.

#Re2.8: Other minor revisions include the following:

-          Some acronyms should be revised. For example, in line 42-43, "small cell neuroendocrine carcinomas (NECs)" could be replaced with "small cell neuroendocrine carcinomas (SCLC)" and "large cell NECs" with "large cell NECs (LCLC)"; contrast-enhanced CT (line 142).

#Re2.8: Thanks for noticing this issue, we have modified the text accordingly.

#Re2.9: Lines 406-407, please rephrase.

#Re2.9: Thanks for this comment. We have rephrased the paragraph as written above.

#Re2.10: Lines 344-346-347, correct "FDG" to "[18F]F-FDG".

#Re2.10: Thanks for noticing this issue, we have modified the text accordingly.

Reviewer 3 Report

Comments and Suggestions for Authors

The fellow researchers have produced an excellent review work, based largely on the classification, which is absolutely appreciated, and on the use of imaging for the diagnosis and control of this pathology which unfortunately is increasingly frequent in the population and if up to 15 years ago it affected after the age of 60 we now find it at a much younger age. We agree with everything that has been written but we like to make some clarifications. The first concerns the therapeutic approach with somatostatin and analogues which can cause complications (PMID: 38051513 to be cited in the bibliography) the second the everolimus which with the inhibition of calcineurin is a protector of renal failure caused by other drugs especially if the patients have been transplanted and we know they are more exposed to cancer. The last concerns gallium PET which represents one of the key points in the diagnosis and subsequently in the follow-up if we are faced with metastatic neuroendocrine tumors. English to be reviewed, excellent bibliography and absolutely sufficient for the theses presented, good iconography.

Comments on the Quality of English Language

English needs to be revised

Author Response

#Re3.1: The first concerns the therapeutic approach with somatostatin and analogues which can cause complications (PMID: 38051513 to be cited in the bibliography)

#Re3.1 Thanks for noticing this, we have added this reference to the text and modified the text as follows: “However, SSAs may disrupt gallbladder motility by affecting enzyme cascades, leading to the formation of gallstones. Therefore, it is recommended considering prophylactic cholecystectomy during the initial surgery for NETs to prevent future emergency surgeries for gallstone complications”.

#Re3.2: the second the everolimus which with the inhibition of calcineurin is a protector of renal failure caused by other drugs especially if the patients have been transplanted and we know they are more exposed to cancer.

#Re3.2. Thanks for this suggestion. We have added a sentence regarding this aspect, as follows: “Additionally, everolimus helps protect against renal failure caused by other drugs, which is particularly important for transplant patients who are at a higher risk of developing cancer”.

#Re3.3: The last concerns gallium PET which represents one of the key points in the diagnosis and subsequently in the follow-up if we are faced with metastatic neuroendocrine tumors.

#Re3.3:Thanks for this suggestion. We have added a sentence on the text as follows: “[68Ga]DOTATATE PET/CT is a highly effective imaging technique for diagnosing and locating primary tumors in patients with metastatic NENs, whether of known or unknown origin. It plays a fundamental role in both initial diagnosis and ongoing follow-up”

#Re3.4 English to be reviewed, excellent bibliography and absolutely sufficient for the theses presented, good iconography.

#Re3.4 . Thanks for noticing this. English proofreading has been done.